# Usefulness of an Additional Filter Created Using 3D Printing for Whole-Body X-ray Imaging with a Long-Length Detector

**DOI:** 10.3390/s22114299

**Published:** 2022-06-06

**Authors:** Hyunsoo Seo, Wooyoung Kim, Bongju Han, Huimin Jang, Myeong Seong Yoon, Youngjin Lee

**Affiliations:** 1Department of Radiological Science, College of Health Science, Gachon University, 191, Hambakmoero, Yeonsu-gu, Incheon 21936, Gyeonggi-do, Korea; hyunsooeun99@gmail.com (H.S.); rladndud3207@gmail.com (W.K.); 2Quality Assurance Team, Business Division, Vieworks, 41-3, Burim-ro 170beon-gil, Dongan-gu, Anyang-si 14055, Gyeonggi-do, Korea; radiohan@vieworks.com; 3Department of Radiology, Namcheon Hospital, 575, Gosan-ro, Gunpo-si 15820, Gyeonggi-do, Korea; gmlals815@naver.com; 4Department of Emergency Medicine, College of Medicine, Hanyang University, 222-1, Wangsimni-ro, Seongdong-gu, Seoul 04763, Korea

**Keywords:** long-length detector, additional filter, 3D printing technology, whole spine examination, long leg examination, evaluation of image quality and dose

## Abstract

We recently developed a long-length detector that combines three detectors and successfully acquires whole-body X-ray images. Although the developed detector system can efficiently acquire whole-body images in a short time, it may show problems with diagnostic performance in some areas owing to the use of high-energy X-rays during whole-spine and long-length examinations. In particular, during examinations of relatively thin bones, such as ankles, with a long-length detector, the image quality deteriorates because of an increase in X-ray transmission. An additional filter is primarily used to address this limitation, but this approach imposes a higher load on the X-ray tube to compensate for reductions in the radiation dose and the problem of high manufacturing costs. Thus, in this study, a newly designed additional filter was fabricated using 3D printing technology to improve the applicability of the long-length detector. Whole-spine anterior–posterior (AP), lateral, and long-leg AP X-ray examinations were performed using 3D-printed additional filters composed of 14 mm thick aluminum (Al) or 14 mm thick Al + 1 mm thick copper (Cu) composite material. The signal-to-noise ratio (SNR), contrast-to-noise ratio (CNR), and radiation dose for the acquired X-ray images were evaluated to demonstrate the usefulness of the filters. Under all X-ray inspection conditions, the most effective data were obtained when the composite additional filter based on a 14 mm thick Al + 1 mm thick Cu material was used. We confirmed that an SNR improvement of up to 46%, CNR improvement of 37%, and radiation dose reduction of 90% could be achieved in the X-ray images obtained using the composite additional filter in comparison to the images obtained with no filter. The results proved that the additional filter made with a 3D printer was effective in improving image quality and reducing the radiation dose for X-ray images obtained using a long-length detector.

## 1. Introduction

Scanography is the only method that allows for complete spine evaluations through whole-spine scanography (WSS) and examinations of long bones [1]. In particular, regular WSS examinations until adulthood are necessary for children and adolescents with scoliosis [2]. Image analysis of these patients is performed using methods to characterize the type and severity of the curvature by simultaneously evaluating the Cobb angle and axial rotation of the spine [3]. In addition, identification of the underlying pathological condition of patients with scoliosis and detection of increments in deformity are essential for making treatment decisions [1]. Long bone examinations, which are relatively common in patients with limb length discrepancies (LLDs), are mostly performed in patients with deformities of the femur or tibia. Since anteroposterior long-leg X-ray images are required to accurately analyze these deformities, images that can quantify LLD and limb deformities should be obtained first [4].

During X-ray examinations of patients with scoliosis, images are acquired while standing under the assumption that the patient is able to walk. Under this assumption, images were previously acquired through three X-ray irradiations using a long 35.56 × 91.44 cm^2^ detector covering the entire spine [5]. However, continuing research on the size of the detector by various medical device companies and laboratories led to the development of detector forms that extend to a maximum of 43.18 × 43.18 cm^2^ and are widely used for digital radiography (DR) image acquisition [6]. With the introduction of a detector capable of acquiring DR images, it is now possible to improve image quality, which was not possible with the existing film-screen technology, and to use these advancements to reduce the radiation dose in diagnostic medical examinations [7]. However, because the detectors in such detector systems need to be moved after X-ray imaging of one area during whole-body examinations, problems associated with an increase in examination time and patient movement inevitably occur [8]. To compensate for these shortcomings, our research team recently successfully developed a long-length detector that can acquire whole-body images in one X-ray irradiation by mounting a 43.18 × 129.54 cm^2^ detector in a holder. By using the developed X-ray system with a long-length detector, it may be possible to achieve greater accuracy in patients with limited mobility or pediatric scoliosis while reducing the imaging time. In particular, this system is expected to allow dose reduction by producing lower noise, in addition to yielding higher sensitivity and dynamic range in comparison with the existing methods [9].

Current DR systems synthesize several individual images based on an automatic stitching method to generate a single combined image [8]. The application of this method to obtain X-ray images of an anatomical structure such as a relatively long spine or leg can improve the usefulness of diagnosis. However, when a 43.18 × 43.18 cm^2^ detector is used for spine or leg diagnostic examinations, acquisition of whole-body images with a single X-ray exposure is difficult, necessitating at least three consecutive scans [6]. In contrast, by using the long-length detector developed by our research team, a single image of a long part of the human body can be obtained, thereby reducing the re-exposure rate and radiation dose to the patient. Moreover, the long-length detector can reduce various artifacts generated while merging several X-ray images.

However, X-ray examinations of the whole body or elongated parts of the human body are associated with various problems because these examinations are performed using energy settings suitable for relatively thick parts. As a result, for relatively thin bones, such as the cervical spine (C-spine) or ankle, the X-rays show greater penetration than before, making it difficult to obtain a clear image. To solve this problem, an additional filter using aluminum (Al) or copper (Cu) material has been used, which yielded improved X-ray image quality while simultaneously reducing the dose [10,11].

According to Morishima et al., in X-ray-based videofluoroscopic swallowing studies, a dose reduction effect of 15.4% to 55.1% can be achieved when an additional Al filter is used while maintaining image quality [10]. In addition, Kawashima et al. demonstrated that when an additional Cu filter was applied to general abdominal radiography, the entrance surface dose could be reduced by approximately 30% or more in comparison with the case where no filter was applied [11]. In particular, the degree of dose reduction was confirmed to improve the contrast by approximately 20% to 40%.

However, the use of an additional filter in X-ray imaging systems is inevitably associated with an increased X-ray tube load to compensate for the decrease in radiation dose [11]. Another limitation is the cost increase as a result of the use of an additional filter, and 3D printing technology has been proposed as a method to address these shortcomings. Three-dimensional printing technology involves manufacturing objects in 3D form by fusing or depositing materials, such as plastics, metals, and powders, in layers [12]. Typically, additive manufacturing (AM) technology enables the custom fabrication of 3D structures using computer-aided design (CAD) software [13]. Three-dimensional printing technology has been developing rapidly and is most widely applied in the medical field, and the printed products have become cheaper and more accessible to users [13,14]. These advancements in 3D printing technology offer advantages for both medical professionals and patients [15]. Using 3D printers, customized medical products and equipment can be freely produced, and items necessary for diagnostic assistance can be manufactured quickly and inexpensively [12]. In addition, this approach can be applied to various situations only by changing the 3D printing technology and materials used [16]. Thus, 3D printing can complement medical practice, and many technological contributions and developments based on 3D printing can be expected to contribute to the medical field in the future [14].

The additional filter manufactured using this 3D printing technology is expected to provide various advantages in X-ray imaging using the long-length detector developed by our research team. Thus, the purpose of this study was to obtain X-ray images using additional Al and Cu filters manufactured using 3D printing technology based on a long-length detector and confirm the applicability of this technique for whole-spine and long-leg imaging. For the purpose of this study, the X-ray image quality and radiation dose with and without additional filters were quantitatively evaluated and compared.

## 2. Materials and Methods

Whole-spine anteroposterior (AP), lateral (LAT), and long-leg AP images were obtained using a PBU-60 Rando phantom (KYOTO KAGAKU, Kyoto, JAPAN), which has a tissue density similar to that of the human body. The equipment used to acquire the X-ray images was GR10X-40K (VIEWORKS, Anyang, Korea), and the detectors used were the FXRD-4386W (VIEWORKS, Anyang, Korea) and FXRD-1717N (VIEWORKS, Anyang, Korea) models. The overall dimensions of the FXRD-4386W and FXRD-1717N detectors were 43.18 × 86.36 cm^2^ and 43.18 × 43.18 cm^2^, respectively. GR10X-40K, a long-length detector of 43.18 × 129.54 cm^2^, was manufactured by attaching two detectors together. The phantom and long-length detector used in the experiment are shown in Figure 1. Table 1 lists the parameters and time required for the X-ray image acquisition.

The 3D printer used in this study is shown in Figure 2a. The filter design was manufactured as shown in Figure 2b based on the 3D human body shape provided by the National Institute of Technology and standards information from the Ministry of Trade, Industry and Energy of Korea using the CAD (Solidworks software, DASSAULT, Hollywood, FL, USA) program. After transmitting the information of the designed filter to the 3D printer, a 14 mm thick Al filter and a 14 mm thick Al + 1 mm thick Cu composite filter were manufactured using a filament. Figure 2c,d show the 14 mm thick Al whole-spine and long-leg filters, respectively. The anterior view of the 14 mm thick Al + 1 mm thick Cu composite filter is presented in Figure 2e. Figure 2f,g show the 14 mm thick Al + 1 mm thick Cu composite whole-spine and long-leg filters, respectively. Figure 2h shows the manufactured filter installed for use in the experiments.

Images were acquired using Image J software (version 2.0; National Institutes of Health, Bethesda, MD, USA), with 5 × 5 pixels of ROI_target_ and ROI_background_. An image with a high diagnostic value was obtained by measuring and analyzing the signal-to-noise ratio (SNR) and contrast-to-noise ratio (CNR), which are quantitative methods for the evaluation of noise. As shown in Figure 3, ROI_target_ and ROI_background_ were selected for the C-3, thoracic spine (T-6), and lumbar spine (L-3) parts of the whole-spine AP image. Under the same conditions, the ROIs of C-3, T-6, and L-3 of the whole-spine LAT image were selected as shown in Figure 4, and the pelvis, knee, and ankle ROIs of the long-leg AP image were selected as shown in Figure 5.

The SNR equations of the set ROI_target_ for quantitative evaluation are as follows:(1)$$\mathrm{SNR}=\frac{S_{A}}{\sigma_{A}}$$
where $S_{A}$ is the average value of the signal intensity in ROI_target_ and $\sigma_{A}\;$ is the standard deviation of the signal intensity in ROI_target_. As the SNR value increases, the signal becomes higher than the amount of noise, so the image can be confirmed to be of good quality.

The CNR equations for the set ROI_target_ and ROI_background_ for quantitative evaluation are as follows:(2)$$\mathrm{CNR}=\frac{\left|S_{A}-S_{B}\right|}{\sqrt{\sigma^{2}{}_{A}+\sigma^{2}{}_{B}}}$$
where $S_{A}$ and $S_{B}$ represent the average values of the signal intensity in the ROI_target_ and ROI_background_, respectively, and $\sigma_{A}$ and $\sigma_{B}$ represent the standard deviation of the signal intensity in the ROI_target_ and ROI_background_, respectively. A higher CNR value can be confirmed to indicate better image quality.

## 3. Results

### 3.1. X-ray Images and Quantitative Evaluations

Figure 6, Figure 7 and Figure 8 show the acquired whole-spine AP, LAT, and long-leg AP X-ray images with and without additional filters, respectively. Visual evaluation of the obtained images confirmed that the image quality was improved when an additional filter based on Al and Cu materials manufactured with a 3D printer was used in thin areas such as the C-spine and ankle. In particular, the highest quality was observed in all the acquired X-ray images when a filter combining 14 mm thick Al + 1 mm thick Cu was used.

Table 2 and Figure 9 show the SNR and CNR values and graphs for each measurement area according to the examination.

In the whole-spine AP examinations of the C-spine, T-spine, and L-spine areas, the SNRs with the 14 mm thick Al + 1 mm thick Cu composite additional filter were, respectively, 63%, 35%, and 29% better than those obtained using no additional filter, while the CNRs were, respectively, 43%, 35%, and 20% better than those obtained using no additional filter.

In whole-spine LAT examinations of the C-spine, T-spine, and L-spine areas, the SNRs with the 14 mm thick Al + 1 mm thick Cu composite additional filter were, respectively, 69%, 43%, and 50% better than those obtained using no additional filters, while the CNRs were, respectively, 43%, 28%, and 32% better than those obtained with no additional filter.

Lastly, in long-leg AP examinations of the pelvis, knee, and ankle areas, the SNRs with the 14 mm thick Al + 1 mm thick Cu composite additional filter were, respectively, 29%, 16%, and 81% better than those obtained no additional filter, while the CNRs were, respectively, 38%, 37%, and 58% better than those obtained with no additional filter.

### 3.2. Radiation Dose

The PCXMC program was used for dosimetry of the X-ray imaging system used in this study. This program is a Monte Carlo-based software and is well known for its high accuracy in effective dose evaluation.

The most effective radiation dose reduction effect was obtained with the 14 mm thick Al + 1 mm thick Cu composite additional filter. The radiation doses in the whole-spine AP, LAT, and long-leg AP examinations with the composite additional filter showed 91%, 88%, and 90% reduction in comparison with the doses in the examinations with no additional filter. Similarly, the radiation doses in the whole-spine AP, LAT, and long-leg AP examinations with the composite additional filter showed 79%, 75%, and 81% dose reduction in comparison with the examinations using a 14 mm thick Al additional filter. The radiation doses were significantly lower when using the composite filter in comparison with other cases, as presented in Table 3 and Figure 10.

## 4. Discussion

Whole-spine and long-leg examinations are currently performed using digital radiography (DR) detectors [7]. However, because the detector needs to be moved for X-ray imaging of different areas during whole-body examinations based on such detector systems, problems caused by increased examination time and patient movement inevitably occur [8]. Over repeated radiographic examinations, radiosensitive organs such as the thyroid gland or gonads are exposed to high cumulative effective doses [1]. In addition, the large dynamic range may result in unintentionally high radiation doses [17]. Currently, most detectors are up to 43.18 × 43.18 cm^2^ in size, and several shots and image-stitching techniques are needed for whole-spine or long-leg examinations [9]. Thus, although DR systems can improve image quality and produce low-dose radiation effects in comparison with the existing screen-film systems, the radiation dose problem requires careful consideration [18].

To compensate for these shortcomings, our research team developed a long-length detector that can acquire a whole-body image with a single X-ray irradiation by mounting a 43.18 × 129.54 cm^2^ detector in a holder. With this approach, a more comprehensive analysis can be performed by visualizing the entire spine and long leg, and the risk of measurement errors due to radiation exposure and patient movement and the risk of falling for patients with reduced mobility are reduced because only one irradiation is required. However, examination of long parts of the human body or the whole body with X-rays requires high-energy X-rays to ensure high image quality; conversely, these X-rays show greater penetration in relatively thin bones, such as the C-spine or ankle, making it difficult to obtain a clear image.

To compensate for these shortcomings, aluminum and copper filters have been fabricated using 3D printing technology. AM technology can enable the custom fabrication of 3D structures using CAD software [13]. The development of an additional filter accounting for the anatomical structure of the whole spine and long leg can result in image-quality improvement and radiation dose-reduction effect. An aluminum filter with a thickness of at least 2.5 mm is recommended for most radiation facilities [19]. Aluminum filters are used in X-ray filtration to reduce the number of low-energy X-ray photons reaching the patient, thereby reducing the radiation dose [20]. Some low-energy radiation is completely absorbed by the patient and not used to generate X-ray images, unnecessarily increasing the dose to the patient. A thin metal plate, such as copper or aluminum, can be used as an additional filter to harden the photon beam, and the low-energy ratio can be reduced through additional filtration. Some authors recommend using an additional filter instead of reducing the kVp to reduce the patient’s radiation dose [19]. The copper filter absorbs low-energy X-rays that do not contribute to imaging, improves the quality of the X-ray image, and simultaneously reduces the dose [11]. According to Samai et al., additional copper filters can provide optimal conditions for digital X-ray imaging of the chest [21]. Moreover, Martin suggested that this approach was effective enough to maintain image quality while reducing the entrance surface dose (ESD) by 40%–50% for abdominal imaging [11]. The use of 0.1 mm and 0.2 mm thick copper filters has been suggested and is commonly used in pediatric radiography [19]. Since copper absorbs approximately 25 times more low-energy X-rays than aluminum filters, we fabricated an additional 14 mm thick Al + 1 mm thick Cu composite filter.

The results confirmed that both the SNR and CNR values were higher when using the composite additional filter in comparison with the findings obtained when no filter was used. The SNR data for whole-spine and long-leg examinations showed that the images acquired for the C-spine, T-spine, L-spine, pelvis, knee, and ankle areas with the composite additional filter were, respectively, 66%, 39%, 40%, 29%, 16%, and 81% better than the corresponding images with no additional filter. The CNR data for whole-spine and long-leg examinations showed that the images acquired for the C-spine, T-spine, and L-spine, pelvis, knee, and ankle areas with a 14 mm thick Al + 1 mm thick Cu composite additional filter were, respectively, 43%, 32%, 26%, 38%, 37%, and 58% better than those obtained using no additional filter. The usefulness of the additional filter was proven by the fact that the largest increase in SNR and CNR values was obtained in examinations of thin bones such as the C-spine and ankle. In the results for radiation dose, whole-spine and long-leg examinations showed 90% reductions in the radiation dose when using a 14 mm thick Al + 1 mm thick Cu composite additional filter, in comparison with the radiation dose for examinations with no additional filter. Thus, the use of an additional filter fabricated with a 3D printer was effective for both improving the image quality and reducing the radiation dose.

To reduce the patient’s radiation dose, it is essential to consider the detector features and final image quality of the whole-spine and long-leg examinations. To maintain the image quality required for accurate medical diagnosis, radiation dose minimization must be optimized. Because the detector is the main component of a DR system, it must satisfy the following requirements: (1) The field size must be sufficiently large for all radiographic applications, and the pixel size must be sufficiently small to allow sufficient resolution. (2) The sensitivity must be sufficiently high to allow low-dose operation, and the dynamic range must be sufficiently large to cover a wide range of intensities. (3) In addition, readout times must be fast, and internal noise sources must be sufficiently small to preserve image quality [9]. The use of a long-length detector that satisfies all of these conditions and the addition of copper and aluminum filters fabricated with a 3D printer can be a new method for whole-spine and long-leg examinations.

This study had several limitations. The first was the use of an adult standard phantom, which had tissue density similar to that of the human body. This approach necessitated considerations for various physical differences in children and adults with obesity [22]. Moreover, the phantom did not account for pathological factors, such as lesions in internal organs, which can be another limitation since the results would be different from those obtained in actual lesion evaluations. The wide and arbitrary range of experimental conditions, such as tube voltage, tube current, thickness of the additional filter, and ROI setting location, can also be limiting factors [3]. Future studies should consider analyzing actual patient data and applying actual clinical conditions for a more realistic evaluation, and additional studies on the fabrication of additional filters based on the thickness of the patient’s body part are also needed [17,22].

## 5. Conclusions

The purpose of this study was to improve the image quality and reduce the patient radiation dose during whole-spine and long-leg examinations with a long-length detector by using additional aluminum and copper filters fabricated with 3D printing technology. With the additional filter, the SNR and CNR values were higher and the radiation dose was lower than the corresponding values obtained with no additional filter, confirming that these filters were useful for image diagnosis. The results confirmed that the image acquisition time could be shortened compared to that with the existing three shots during a single examination using a long-length detector. In addition, the superior image quality obtained with the additional filter demonstrated that images of high diagnostic value could be created in whole-spine and long-leg examinations.

## Figures and Tables

**Figure 1 sensors-22-04299-f001:**
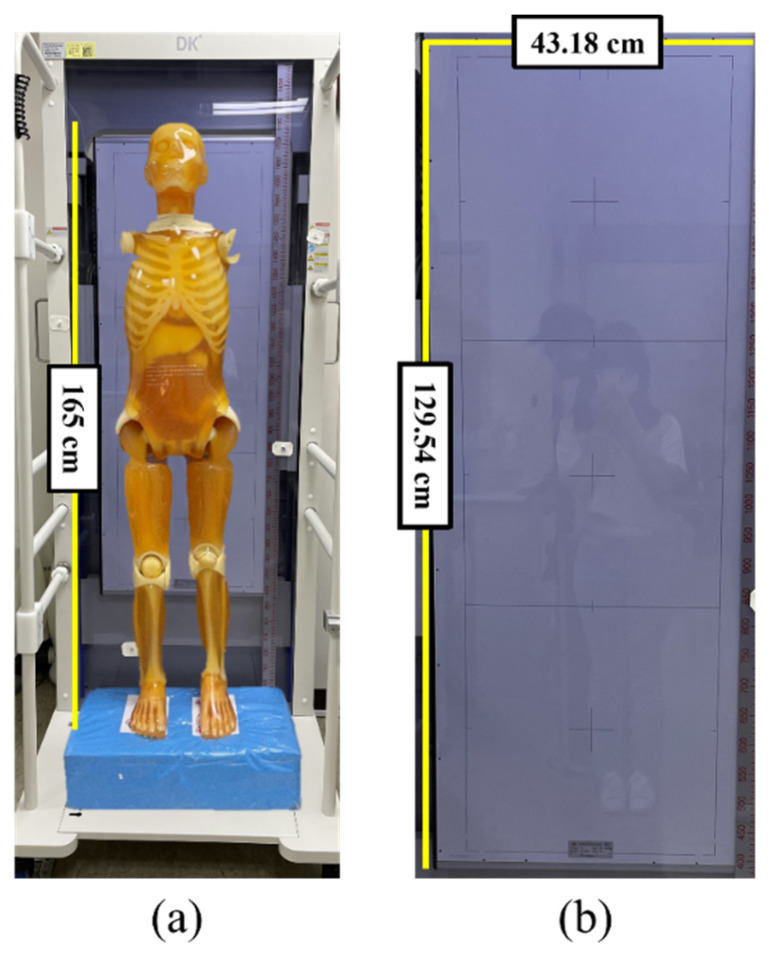
(**a**) Whole-body phantom and (**b**) long-length detector used in the study.

**Figure 2 sensors-22-04299-f002:**
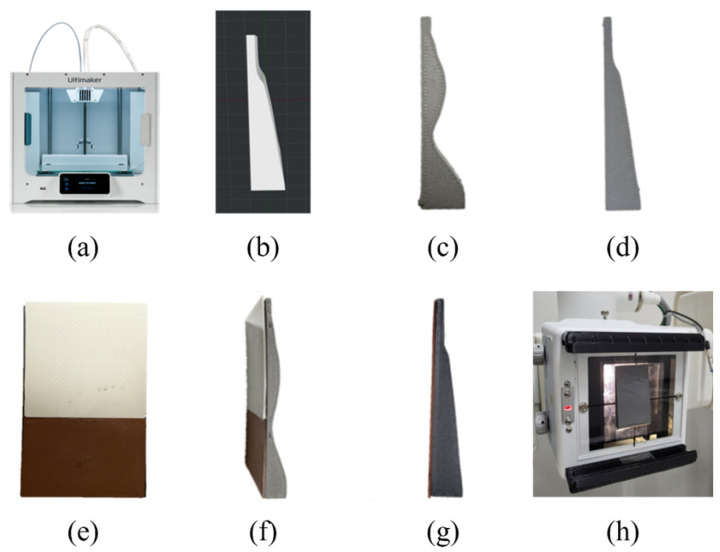
Material and equipment used in the experiment: (**a**) using the Ultimaker 3D printer; (**b**) long-leg filter 3D modeling using the CAD program; (**c**) side view of the completed 14 mm thick Al whole-spine filter; (**d**) side view of the completed 14 mm thick Al long-leg filter; (**e**) anterior view of the completed 14 mm thick Al + 1 mm thick Cu composite whole-spine filter; (**f**) side view of the completed 14 mm thick Al + 1 mm thick Cu composite whole-spine filter; (**g**) side view of the completed 14 mm thick Al + 1 mm thick Cu composite long-leg filter; (**h**) filter installed for use in experiments.

**Figure 3 sensors-22-04299-f003:**
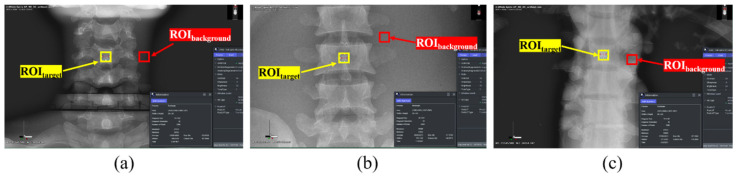
ROI setting in the whole-spine AP image: (**a**) ROI_target_ and ROI_background_ of C-3; (**b**) ROI_target_ and ROI_background_ of T-6; (**c**) ROI_target_ and ROI_background_ of L-3.

**Figure 4 sensors-22-04299-f004:**
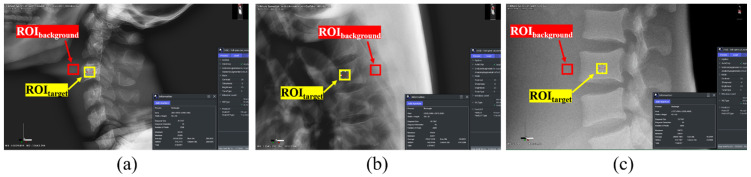
ROI setting in the whole-spine LAT image: (**a**) ROI_target_ and ROI_background_ of C-3; (**b**) ROI_target_ and ROI_background_ of T-6; (**c**) ROI_target_ and ROI_background_ of L-3.

**Figure 5 sensors-22-04299-f005:**
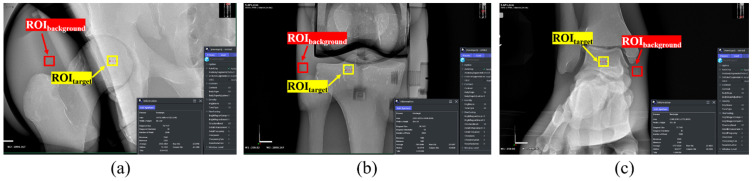
ROI setting in the long-leg AP image: (**a**) ROI_target_ and ROI_background_ of the pelvis; (**b**) ROI_target_ and ROI_background_ of the knee; (**c**) ROI_target_ and ROI_background_ of the ankle.

**Figure 6 sensors-22-04299-f006:**
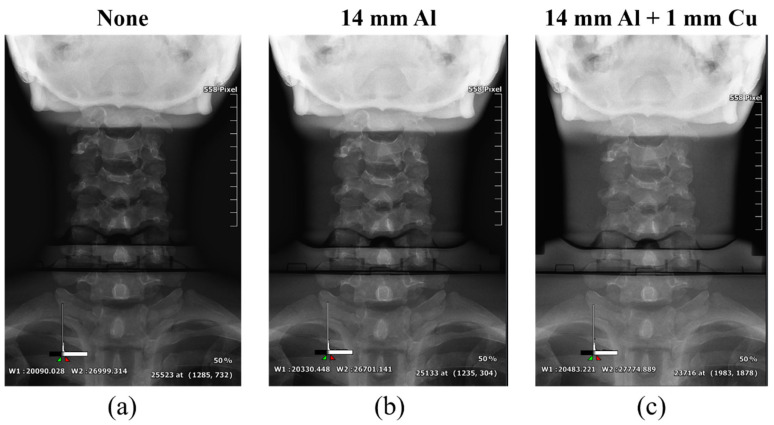
Acquired whole-spine AP X-ray images without and with additional filters. (**a**) Whole-spine AP image (**a**) without an additional filter, (**b**) with a 14 mm thick Al additional filter, and (**c**) with a 14 mm thick Al + 1 mm thick Cu composite additional filter.

**Figure 7 sensors-22-04299-f007:**
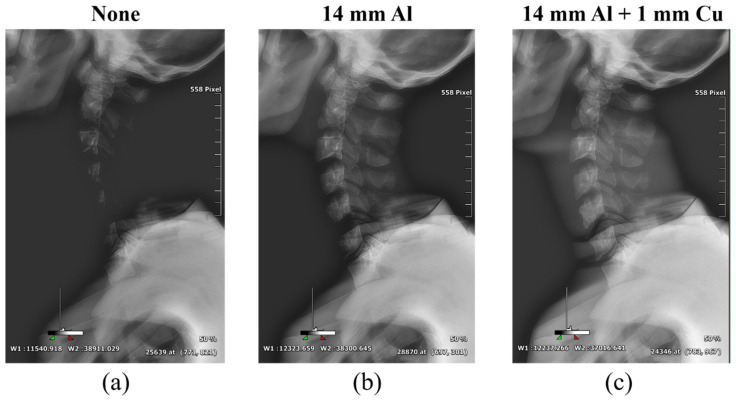
Acquired whole-spine LAT X-ray images obtained without and with additional filters. (**a**) Whole-spine LAT image (**a**) without an additional filter, (**b**) with a 14 mm thick Al additional filter, and (**c**) with a 14 mm thick Al + 1 mm thick Cu composite additional filter.

**Figure 8 sensors-22-04299-f008:**
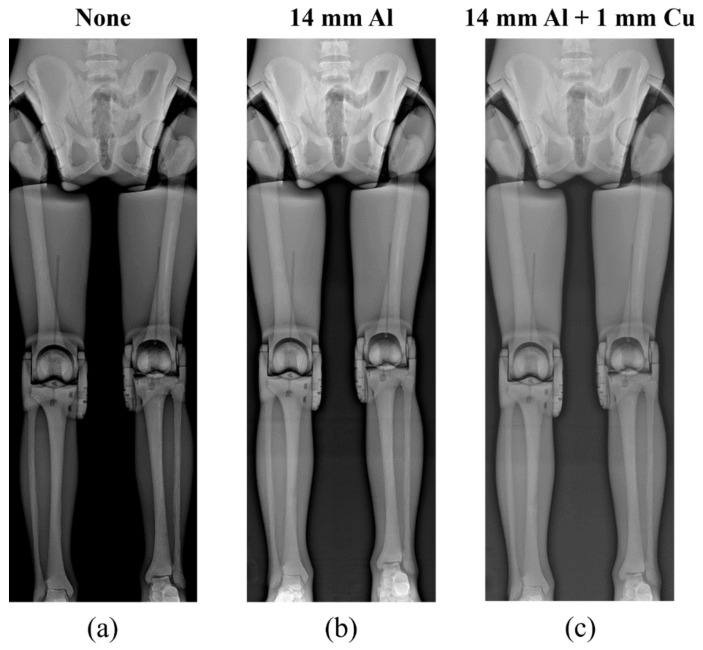
Acquired long-leg AP X-ray images without and with additional filters. (**a**) Long-leg AP image (**a**) without an additional filter, (**b**) with a 14 mm thick Al additional filter, and (**c**) with a 14 mm thick Al + 1 mm thick Cu composite additional filter.

**Figure 9 sensors-22-04299-f009:**
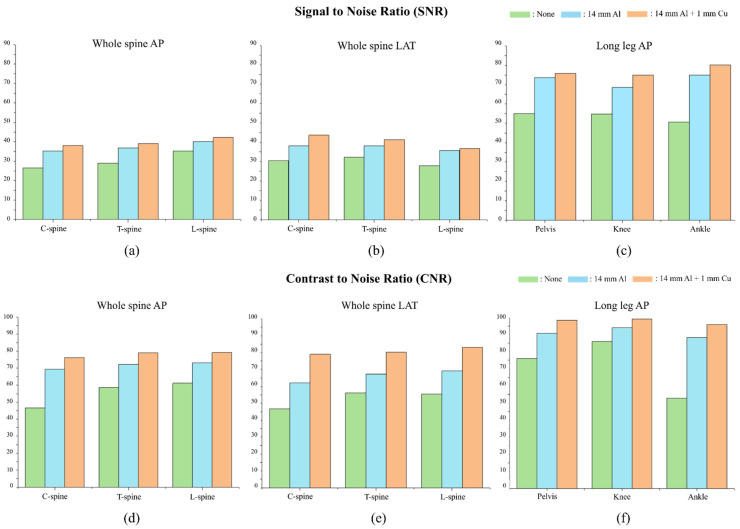
Graphs of the SNR and CNR results evaluated with and without additional filters. SNR results for (**a**) whole-spine AP, (**b**) whole-spine LAT, and (**c**) long-leg AP examinations. CNR results for (**d**) whole-spine AP, (**e**) whole-spine LAT, and (**f**) long-leg AP examinations.

**Figure 10 sensors-22-04299-f010:**
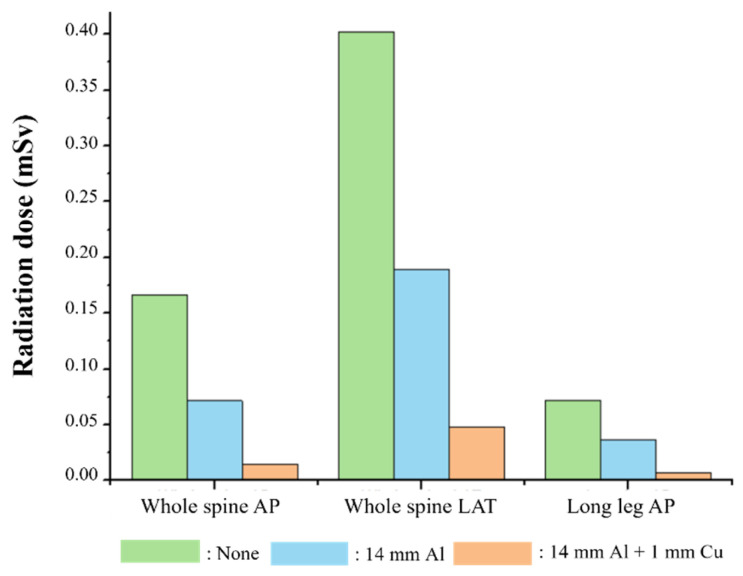
Graph of the evaluated radiation dose (mSv) in examinations with and without additional filters.

**Table 1 sensors-22-04299-t001:** Set X-ray image examination type, exposure conditions, and image acquisition time.

Type of X-ray Examination	Examination Area	Source-to-Image Distance (SID)	kVp	mAs	Image Acquisition Time
Whole-spine AP	C-spine	250 cm	90	50	10.02 s
	T-spine				
	L-spine				
Whole-spine LAT	C-spine	250 cm	105	80	10.06 s
	T-spine				
	L-spine				
Long-leg AP	Pelvis	250 cm	90	50	9.7 s
	Knee				
	Ankle				

**Table 2 sensors-22-04299-t002:** SNR and CNR results with respect to the examination method.

		SNR			CNR		
Type of X-ray Examination	Examination Area	None	14 mm Al	14 mm Al + 1 mm Cu	None	14 mm Al	14 mm Al + 1 mm Cu
Whole-spine AP	C-spine	46.72	69.39	76.17	26.53	35.19	38.07
	T-spine	58.76	72.12	79.08	29.07	36.88	39.15
	L-spine	61.16	73.08	79.11	35.17	40.11	42.21
Whole-spine LAT	C-spine	46.80	61.96	78.93	30.44	38.04	43.65
	T-spine	56.17	67.18	80.17	32.30	38.10	41.38
	L-spine	55.51	69.04	83.15	27.93	35.66	36.76
Long-leg AP	Pelvis	76.10	90.76	98.32	55.02	73.68	75.85
	Knee	85.79	93.91	99.11	54.79	68.68	75.01
	Ankle	53.03	88.27	95.83	50.68	74.95	80.18

**Table 3 sensors-22-04299-t003:** Evaluated radiation doses (mSv).

Type of X-ray Examination	None	14 mm Al	14 mm Al + 1 mm Cu
Whole-spine AP	0.1665	0.0715	0.0150
Whole-spine LAT	0.4017	0.1896	0.0477
Long-leg AP	0.0724	0.0362	0.0069

## Data Availability

Not applicable.

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
