# Peer review of "Usefulness of an Additional Filter Created Using 3D Printing for Whole-Body X-ray Imaging with a Long-Length Detector"

_sensors, 2022, doi:10.3390/s22114299_

Round 1

Reviewer 1 Report

This paper developed the filter for X-ray imaging using 3D printing and presented the usefulness. The title should be changed. 

"Usefulness of an additional filter created using 3D printing technology for whole-body X-ray imaging  with a self-developed long-length detector" 

Author Response

Thank you for review and comment in this manuscript.
We have revised the paper as your suggestion and responded point by point.
Please confirm attached revised manuscript and response files.

Best regards,

Youngjin Lee

Reviewer 2 Report

Please address my comments listed in the attached file.

Author Response

(The authors gave the same response as above.)

Reviewer 3 Report

This is an excellently written paper. The project is focused on improving quality and reducing radiation dose in a whole-spine and long-leg examination of a patient. The experimental results showed that these goals were accomplished. I have just one suggestion:

The dimension of the long-length detector is 17 × 51 inches as stated in Line 130 on page 3. I suggest adding this dimension to Figure 1b, and also adding the dimension or height of the whole-body phantom to Figure 1a.

Author Response

(The authors gave the same response as above.)

Reviewer 4 Report

The authors describe how to improve image quality and lower radiation dose for X-ray images by using a long-length detector as part of their research. The use of 3D printed filters to enhance the applicability of the self-developed long-length detector is impressive. These research results suggest good prospects for diagnostic imaging. Although it needs further research to be implemented in clinical practice, this preliminary analysis indicates that 3d printed filters could be effective in actual clinical conditions. It is recommended for publication.

Author Response

(The authors gave the same response as above.)
